# Peripheral Hemodynamics Estimation Using the Photoplethysmography Method

**DOI:** 10.3390/s24247882

**Published:** 2024-12-10

**Authors:** Toru Shimuta, Kaname Hanada, Kazuteru Ryu, Koichi Idei, Nobuyuki Kanai

**Affiliations:** 1Minato MIRAI Innovation Center, Murata Manufacturing Co., Ltd., 4-3-8, Minatomirai, Nishi-ku, Yokohama-shi 220-0012, Japan; khanada@murata.com; 2Orthopedic Surgery, Kanai Hospital, 612-12, Yodokizu-cho, Fushimi-ku, Kyoto 613-0911, Japan; dr.ryu@kanaihospital.jp (K.R.); idei@koto.kpu-m.ac.jp (K.I.); 3Internal Medicine, Kanai Hospital, 612-12, Yodokizu-cho, Fushimi-ku, Kyoto 613-0911, Japan; dr.kanai@kanaihospital.jp

**Keywords:** peripheral hemodynamics, photoplethysmography device, ring-wearable device, finger photoplethysmogram, photoplethysmography method

## Abstract

Diabetes is known to reduce blood circulation in capillaries and arterioles; however, no devices can easily measure this on a daily basis. In this study, we developed a tool for measuring finger photoplethysmograms using green light and near-infrared LEDs. Thereafter, photoplethysmography was conducted on 25 inpatients/outpatients with diabetes and 21 adult males and females who had not been diagnosed with or treated for diabetes, hypertension, or cardiovascular disease (control group). In patients with diabetes, the inverse full width at half-maximum velocity plethysmogram tended to be smaller than that in the control group, and the delay in the green light a-wave peak relative to the near-infrared light a-wave peak in the acceleration plethysmogram was significantly increased. The results suggest that peripheral hemodynamics can be easily estimated at home using a photoplethysmography device mounted on a ring-wearable device.

## 1. Introduction

Blood flow transports nutrients, hormones, metabolic waste products, O_2_, and CO_2_ throughout the body and plays various other roles, including controlling body temperature, pH, and osmotic pressure. Stagnation of blood flow in small blood vessels can be caused by numerous factors, including lack of exercise, a high-fat/high-calorie diet, smoking, fatigue, lack of sleep, and cold environments; diseases, such as diabetes, are also a cause. Moreover, neuropathy, retinopathy, nephropathy, and foot lesions are serious complications of diabetes [1]; therefore, assessing arterial function is important because it is associated with various diseases and treatments [2,3], several of which have been previously used [4,5]. However, while useful for treating inpatients, these methods are unsuitable for monitoring outpatients at home because of their invasiveness, need for expensive equipment, and limitations in daily life. For example, video microscopy [6,7] and ultrasound microvessel imaging [8] are challenging for patients owing to their cumbersome or restrictive nature. In addition, they require skilled operators to analyze the acquired images. Consequently, the application of deep learning has been investigated to address these problems [9]. Other methods include noncontact imaging [10,11] (which cannot be performed without resting in the area where the camera is installed), angiography [12,13] (which is highly invasive), and laser Doppler flowmetry (which does not distinguish between blood flow in the arterioles and venules [6] and provides only relative indices of microvascular perfusion; therefore, it is often used in conjunction with the vasoreactivity test [14]).

Based on the circulatory dynamics of blood in large arteries, devices that can easily measure a patient’s heart rate, blood pressure, etc. have been widely used. If a technology that can easily measure the hemodynamics of arterioles and capillaries could be developed, it would be convenient for use at home. Photoplethysmography (PPG) is a long-standing technology that uses the high absorbance of hemoglobin in the blood to detect changes in blood volume by measuring changes in the amount of light transmitted or reflected through the skin. Moreover, it is noninvasive and inexpensive, and attempts have been made to use it for diagnosis [15,16].

In this study, we developed a simple method for estimating the hemodynamics of arterioles and capillaries using photoplethysmography. We found clear differences in two features—that is, the inverse full width at half-maximum velocity plethysmogram (1/FWHM_VPG_) and the time delay between the green light and the near-infrared *a*-wave peaks—between patients with diabetes and healthy subjects, where differences in microvascular hemodynamics can be assumed.

## 2. Materials and Methods

Because the photoplethysmography sensor noise increases when the device does not adhere to the skin, we manufactured a measurement tool that can be affixed to the finger. The measurement tool was used to obtain photoplethysmograms from the subjects under multiple measurement conditions with varying peripheral hemodynamics. The wrist blood pressure was also measured under each condition. In addition, a program was developed to calculate the plurality of feature values from the photoplethysmogram processed in this study. The experimental method was as follows.

Assuming a ring-shaped device, we fabricated a finger-wearable measuring tool equipped with a photoplethysmography sensor at the base of the finger (Figure 1). The sensor was pressed onto the finger using elastic silicone rubber to ensure that the photoplethysmography sensor adhered to fingers of different thicknesses. The photoplethysmography sensor was equipped with three LEDs as light sources, and the photoplethysmograms were measured at three wavelengths: green (approximately 525 nm), red (approximately 660 nm), and near-infrared (approximately 940 nm). A red LED was used to calculate the oxygen saturation level. The tool was equipped with a photodiode (PD) as a light receiver, and three LEDs were sequentially emitted in a time-division manner to irradiate the skin of the finger. The light is received by the PD, and it is converted from analog to digital for analysis. The LED–PD distance was approximately 2.4 mm for the green light and 8.9 mm for the red and near-infrared light. Green light is strongly absorbed by the body, and if the LED–PD distance is long (e.g., 5 mm), it does not reach the PD. In contrast, near-infrared light with a wavelength of 940 nm is used in pulse oximeters and is only weakly absorbed by the body. If the LED–PD distance is as short as 3 mm—for example, the perfusion index, which is the ratio of the pulsatile component to the non-pulsatile component—it becomes difficult to obtain photoplethysmogram waveforms with a high signal-to-noise ratio.

A wrist-cuff blood pressure monitor (HEM-6233T, OMRON Healthcare Co., Ltd., Kyoto, Japan) was attached to the left wrist, and the proposed measurement tool was attached to the index finger of the left hand. Subsequently, in a resting seated position, the left hand (to which the measurement tool was attached) was held at navel, chest, and forehead heights, where photoplethysmograms and blood pressure measurements were taken (Figure 2). Subjects who experienced difficulty raising their left hand to their forehead underwent the same measurement using their right hand. The blood flow in the finger was blocked by the cuff, preventing simultaneous photoplethysmogram measurements. Therefore, blood pressure was measured after completing the photoplethysmogram measurements.

The left elbow was then cooled using a cooling agent, while the left hand was held at chest height. Photoplethysmography and blood pressure measurements were performed after cooling for a minimum of 5 min.

The three measured photoplethysmograms were processed using the program developed in this study. First, only the necessary frequency bands were passed through a band-pass filter with lower and higher cutoff frequencies of 0.5 and 20 Hz, respectively. Thereafter, the filtered photoplethysmograms were differentiated and subjected to moving-average processing to obtain the velocity plethysmograms. The velocity plethysmograms were further differentiated, and the acceleration plethysmograms were obtained by performing moving-average processing, after which each was divided into beats.

Figure 3 illustrates the photoplethysmogram height (*S*) and acceleration plethysmogram *a*-wave peak height (*a*); the solid line represents the photoplethysmogram, and the dashed line represents the acceleration plethysmogram. As illustrated in Figure 3, the minimum points of the photoplethysmogram are connected by straight lines, and the height of the maximum point after conducting a slope correction, such that the slope of the straight line is zero, is set as the photoplethysmogram height (*S*).

Figure 4 illustrates the feature values of the velocity and acceleration plethysmograms. Photoplethysmograms are indicated by solid lines, velocity plethysmograms by dotted lines, and acceleration plethysmograms by dashed lines. In this study, the half-width of the maximum peak of the velocity plethysmogram was termed FWHM_VPG_. Acceleration plethysmograms are typically referred to as *a*-, *b*-, *c*-, *d*-, or *e*-waves. The *a*-, *c*-, and *e*-waves were positive peaks, whereas the *b*- and *d*-waves were negative peaks. The difference between the *a*-wave and *b*-wave peak times is called the *ab* time. The signal intensity at the peak apex of the *a*-wave to the *e*-wave can be set as *a-e*, the difference between the *a*-wave and the *b*-wave as *a-b*, and the difference between the *a*-wave and the d-wave as *a-d*. Figure 4 illustrates the photoplethysmogram, velocity plethysmogram, and acceleration plethysmogram normalized such that the maximum peak is 1. The normalized values are referred to as normalized *a-b* and normalized *a-d*.

## 3. Results

### 3.1. Delay Time of Green a-Wave Peak from Near-Infrared a-Wave Peak

The photoplethysmograms of the near-infrared light and green light measured at the same measurement point were second-order differentiated, and their *a*-wave peak times were compared. The results illustrated that a delay occurred in the green *a*-wave peak relative to the near-infrared *a*-wave peak. It is suggested that green light is strongly absorbed by the body and that the LED–PD distance is short (approximately 2.4 mm). Therefore, the information contained in the green light photoplethysmogram was primarily that of the capillaries in the superficial skin area, whereas the near-infrared light was weakly absorbed by the body and reached the deep skin area. Therefore, the information included the arterioles at a depth of approximately 0.3 mm or more. The proposed design increases the LED–PD distance to approximately 8.9 mm for near-infrared light. It would be difficult for information in superficial regions to reach the PD, and the ratio of information in the deep regions would increase. In other words, the time delay between the green light and near-infrared light *a*-wave peaks corresponds to the time required for the pulse wave to reach the superficial region of the skin.

Figure 5 illustrates the optical paths of the green and near-infrared lights received by the PD. Skin tissue is a light scatterer, and LED light incident on the skin spreads in all directions as it is scattered within the skin. The arrows in Figure 5 do not represent the light emitted from the LED but rather the penetration depth of the light received by the PD.

Figure 6 illustrates an example of an acceleration plethysmogram obtained by calculating the time delay between the green light and near-infrared light *a*-wave peaks. The maximum value of the acceleration plethysmogram is the *a*-wave peak, and the length of the arrow in Figure 6 represents the time delay between the green light and near-infrared light *a*-wave peaks. As illustrated in Figure 6, the time delay between the green light and near-infrared light *a*-wave peaks varies considerably during the measurement process. Figure 6a,b show the waveforms when the time delays between the green and near-infrared light *a*-wave peaks are large and small, respectively. When the time delay between the green light and near-infrared light *a*-wave peaks in (a) is long, the amplitude of the green light acceleration pulse is smaller than that in (b), and the waveform shape differs considerably.

Figure 7 compares the time delay between the green and near-infrared light *a*-wave peaks and the reciprocal (1/*S*) of the green light photoplethysmogram amplitude (*S*) in this measurement. It is evident that the time delay between the green and near-infrared light *a*-wave peaks correlates with 1/*S*. A large 1/*S* value (i.e., a small *S*) signifies low blood flow. In this example, it is speculated that 1/*S* increases because of a decrease in the capillary blood flow owing to a temporary decrease in the stroke volume. From this example, it can be speculated that the pulse wave transit time increases as the blood flow in the capillaries decreases. Additionally, the photoplethysmogram amplitude (*S*) varies depending on the contact state and the pressure between the sensor and skin; consequently, it can be difficult to use the absolute value of 1/*S* to estimate the hemodynamics of capillaries because of variations in each measurement.

### 3.2. 1/FWHM_VPG_

Blood pressure in large arteries is typically measured using a cuff-type sphygmomanometer. Blood pressure in blood vessels decreases as blood progresses from arteries to arterioles to capillaries [17], and the blood flow velocity decreases as it progresses from the aorta to the capillaries. The extent of the decrease in blood pressure and blood flow velocity varies depending on the measurement site, individual vascular condition (arteriosclerosis), mental condition (autonomic nerve condition), the environment (temperature and noise), and clothing worn. Moreover, 1/FWHM_VPG_ was related to the hemodynamics in arterioles, particularly in capillaries. The following two aspects were confirmed as features of 1/FWHM_VPG_:
Measurements in adult males and females who had not been diagnosed with or treated for diabetes, hypertension, or cardiovascular disease showed an almost proportional relationship with wrist blood pressure under conditions where the vascular resistance did not change.The value decreased when the blood vessels contracted owing to the cooling in the vicinity of the measurement site. The brachial and wrist blood pressures increased in certain cases.

In addition to the values of features (1) and (2), the following three feature values were obtained:*a*/*S*;(*a-b*)/(*a*-d);1/*ab* time.

Here, *a/S* is the acceleration plethysmogram *a*-wave peak height (*a*) divided by the photoplethysmogram height (*S*) (Figure 3); (*a-b*)/(*a*-d) are the normalized *a-b* divided by the normalized *a-d* (Figure 4); and 1/*ab* time is the difference between the *a*- and *b*-wave peak times (Figure 4). These feature values, including 1/FWHM_VPG_, are related to the sharpness of the increase in the photoplethysmogram waveform.

Figure 8 and Figure 9 illustrate the relationship between wrist systolic blood pressure and each feature value in adult men and women who have not been diagnosed or treated for diabetes, hypertension, or cardiovascular disease. When changing the height of the measurement site (finger) from the heart, the area near the elbow on the side where the finger is located (i.e., the measurement site at chest height) is cooled (plotted for subjects *A*, *B*, and *C*). The orange line denotes subject *A*, blue line denotes subject *B*, and green line denotes subject *C*. Figure 8a shows 1/FWHM_VPG_ (green light), Figure 8b shows *a*/*S* (green light), Figure 8c shows (*a-b*)/(*a*-d) (green light), Figure 8d shows the 1/*ab* time (green light), Figure 9a shows 1/FWHM_VPG_ (near-infrared light), Figure 9b shows *a*/*S* (near-infrared light), Figure 9c shows (*a-b*)/(*a*-d) (near-infrared light), and Figure 9d shows the 1/*ab* time (near-infrared light).

As illustrated in Figure 8a–c, the wrist systolic blood pressure and each feature value tended to be proportional when the height changed (solid line). It is also evident that cooling reduces each feature value and increases the systolic blood pressure of the wrist (dashed line).

Compared with Figure 8a–c, Figure 8d shows that the 1/*ab* time exhibits less dependence on the wrist systolic blood pressure.

Figure 8a–d shows the feature values calculated from the photoplethysmogram measured using the green light LED, and the results of the feature values calculated from the photoplethysmogram measured using the near-infrared light LED are illustrated in Figure 9a–d. The aforementioned trends are unclear, particularly for subject *B* in Figure 9b–d, when compared to those for the green light.

Figure 10 illustrates that the feature values are related to the steep increase in the photoplethysmogram waveform. Examples of waveforms with the same waveform length but different rising slopes in the photoplethysmograms are shown, which were collected from the same subject. Of the two photoplethysmogram waveforms illustrated in Figure 10a, the solid line increases sharply (slope: large). As shown in Figure 10b, the 1/FWHM_VPG_ and *a/S* changed. In Figure 10c, the times (*a-b*)/(*a-d*) and 1/*ab* changed. Therefore, waveforms with large slopes have considerably larger feature values and are related to the sharp increase in the photoplethysmogram waveform. However, for (*a-b*)/(*a-d*), the ratio of “slope: small” to “slope: large” was slightly different from the other feature values. This is because (*a-b*)/(*a-d*) includes the features of the d-wave; consequently, it is a feature value that includes information other than the steepness of the rising slope.

The 1/FWHM_VPG_ value acquired using green light is more applicable to features (1) and (2). It can be speculated that this is because green light has a high bio-absorption rate and is absorbed before reaching the deep skin regions, resulting in only superficial region information being included (i.e., only capillary information). Similarly, near-infrared light includes not only information on capillaries but also information on arterioles; therefore, it can be speculated that it is susceptible to measurement conditions and individual differences.

Of the four feature values, *a/S* and (*a-b*)/(*a-d*) are susceptible to pressure and body-motion noise because the values of *b*, *d*, and *S* are more susceptible to them. Furthermore, (*a-b*)/(*a-d*) contains d-wave information other than the rising slope. Additionally, the 1/*ab* time value tended to be less dependent on the wrist systolic blood pressure. Consequently, it can be speculated that the extent of matching features (1) and (2) is lower for the other feature values than for 1/FWHM_VPG_ (green light).

The following section presents the experimental results, with a focus on two feature values: the time delay between the green and near-infrared light *a*-wave peaks and 1/FWHM_VPG_ (green light).

The use of LEDs or lasers with wavelengths in the vicinity of blue to yellowish-green light (500–550 nm), which are highly absorbed by the body, is suitable for acquiring information on shallow-skin regions. Furthermore, the distance between the light source and the light receiver should be as short as possible, specifically, 1–3 mm.

The measurement sites of the photoplethysmogram were the wrist, neck, face, and ears. Although the fingers were preferred, as the epidermis of the finger is relatively thin, making it easier to measure photoplethysmograms, the paths from the arterioles to the capillaries are less complicated than those of the face, and the values of each feature tended to be stable.

A ring-shaped wearable device equipped with an optical sensor worn on a finger is suitable for measuring photoplethysmograms because there is little discomfort during continuous and intermittent measurements, even when worn for lengthy periods.

### 3.3. Comparison with People with Diabetes

Angiopathy is a well-known complication of diabetes; therefore, patients with diabetes were selected to compare the hemodynamics of the capillaries and arterioles with those of healthy subjects. Patients with diabetes were selected from among inpatients or outpatients at Kanai Hospital between November 2021 and June 2022. The control group consisted of adult males and females who met the following criteria: they had not been diagnosed with or treated for diabetes, hypertension, or cardiovascular disease; they were employees of Murata Manufacturing Co., Ltd.; they satisfied the specified conditions; and they provided consent from May 2021 to July 2021. This study was approved by the Ethics Committee of the Kanai Hospital (approval no. 121). Measurements were conducted indoors at a controlled room temperature to exclude the influence of outside temperatures.

There were 36 subjects (14 males and 22 females) in the group with diabetes and 39 (26 males and 13 females) in the control group. Measurements were performed a minimum of once per person, up to eight times, and the number of acquired data points was 111 for the group with diabetes and 40 for the control group. The control group tended to be younger and had more males than the group with diabetes.

Data that satisfied the following exclusion criteria among these gathered data were excluded from the analysis—that is, when the blood pressure could not be measured using a wrist-cuff-type sphygmomanometer, when the signal-to-noise ratio of the photoplethysmogram signal was less than 200, and when feature values (such as 1/FWHM_VPG_) could not be calculated from the photoplethysmogram signal. This was because when the signal-to-noise ratio of the photoplethysmogram signal was less than 200, the apex of the peak of the acceleration plethysmogram was buried in noise, resulting in a large drop in peak detection accuracy. A confirmed example in which the feature value could not be calculated from a photoplethysmogram signal was one in which body-motion noise was present.

In this study, the signal intensity was defined as the photoplethysmogram height (*S*) in Figure 3, the noise intensity was defined as the root mean square of the remaining components after removing the photoplethysmogram component from the signal, and the ratio of these two values was defined as the signal-to-noise ratio.

The data that remained after the abovementioned exclusion process (50 datasets from 25 patients with diabetes (6 males and 19 females) and 21 datasets from 21 individuals (17 males and 4 females) in the control group) were used for analysis.

Figure 11 shows a graph of the green light 1/FWHM_VPG_ versus the systolic blood pressure measured at the wrist. It is evident that 1/FWHM_VPG_ was concentrated in a smaller range in the group with diabetes than in the control group. Conversely, the systolic blood pressure in the group with diabetes tended to be higher than that in the control group.

The 1/FWHM_VPG_ ranges from 4.1 [1/s] to 9.6 [1/s], with a mean of 5.6 [1/s], in the group with diabetes and from 5.2 [1/s] to 14.1 [1/s], with a mean of 7.9 [1/s], in the control group. The 1/FWHM_VPG_ differed significantly between the group with diabetes and the control group (*p* = 0.00016).

Figure 12 illustrates a graph of the time delay between the green and near-infrared light *a*-wave peaks against the systolic blood pressure measured at the wrist. In the control group, the time delay between the green and near-infrared *a*-wave peaks was concentrated within a small range, whereas in the group with diabetes, the time delay was also widely distributed.

The time delay between the green and near-infrared light *a*-wave peaks ranged from 0.006 [s] to 0.073 [s] with a mean of 0.028 [s] in the group with diabetes and from 0.003 [s] to 0.021 [s] with a mean of 0.012 [s] in the control group. The time delay between the green and near-infrared light *a*-wave peaks differed significantly between the group with diabetes and the control group (*p* = 1.3 × 10^−7^).

Figure 13 illustrates a graph of the time delay between the green and near-infrared light *a*-wave peaks against the green light 1/FWHM_VPG_. In the group with diabetes, the green light 1/FWHM_VPG_ tended to be smaller than that in the control group, and the time delay between the green and near-infrared light *a*-wave peaks tended to be longer.

Figure 14 illustrates a graph of the change in the green light 1/FWHM_VPG_, where the time delay between the green *a*-wave peak and the near-infrared *a*-wave peak when the photoplethysmogram and the blood pressure were measured by holding the left hand with the measurement tool at navel, chest, and forehead height against the wrist-cuff-type sphygmomanometer measurements.

Plotting all the data resulted in considerable data overlap, making it difficult to observe changes; therefore, six datasets were extracted for each of the groups with diabetes and the control group. The extraction criteria were as follows: blood pressure measurements and feature value calculations were conducted at the navel, chest, and forehead heights; wrist blood pressure measurement values were in the descending order of navel height > chest height > forehead height; and those with minimal overlap with other data were extracted.

As illustrated in Figure 14, the green light 1/FWHMVPG shows an approximately proportional positive correlation with wrist blood pressure in the control group. In the group with diabetes, the absolute value of 1/FWHM_VPG_ for green light was low and tended not to be significantly high with respect to wrist blood pressure. It can be speculated that this behavior is due to the following: in the control group, the low vascular resistance from the radial artery to the arterioles and capillaries results in capillary blood pressure following changes in wrist blood pressure, whereas in the group with diabetes, there is vascular resistance. Consequently, even if wrist blood pressure increases, intracapillary blood pressure does not readily increase.

Figure 15 illustrates a graph of the time delay between the green and near-infrared light *a*-wave peaks instead of the 1/FWHM_VPG_ (green light) value for the data plotted in Figure 14. For the same subject, the time delay tended to increase as the wrist blood pressure decreased.

Figure 16 shows a graph of the time delay between the green and near-infrared light *a*-wave peaks with respect to 1/FWHM_VPG_ (green light) for the data plotted in Figure 14. It is evident that a decrease in 1/FWHM_VPG_ (green light) tends to increase the time delay.

## 4. Discussion

It is evident that the 1/FWHM_VPG_ (green light) was small in the group with diabetes, and that the time delay between the green and near-infrared light *a*-wave peaks was large. The suggested mechanism for this is as follows. As the vascular resistance of the capillaries and arterioles increases, the blood pressure in the capillaries and arterioles decreases, making it difficult for blood to flow. Additionally, as the blood pressure decreased, the pulse wave velocity (PWV) decreased; that is, the time delay between the green and near-infrared light *a*-wave peaks increased.

The time delay between the green *a*-wave peak and near-infrared *a*-wave peak varies with the path length. Consequently, the absolute value of the time delay between the near-infrared light *a*-wave and green light *a*-wave peaks varies depending on the mounting position of the measurement tool and individual differences. Thus, it is desirable to determine the peripheral PWV by dividing the path length by the time delay. However, the path lengths of the arterioles and capillaries cannot be measured; therefore, the time delay between the green and near-infrared light *a*-wave peaks can be substituted.

Accurate measurement of blood pressure and blood flow in capillaries can be challenging. Consequently, we conducted the following experiment using 1/FWHM_VPG_ (green light). The measurement tool was designed to contact the finger with appropriate pressure to ensure measurement stabilization. In the experiment, when measuring for 30 s, the finger was pressed strongly against the tool for 10–20 s from the start of the measurement process to increase the pressure, and the change in 1/FWHM_VPG_ (green light) was examined (Figure 17).

It is evident from Figure 17 that the photoplethysmogram direct-current (DC) components of the green light and 1/FWHM_VPG_ (green light) increased when excessive pressure was applied (10–20 s). An increase in the DC component of the photoplethysmogram indicates that the absorption of light by the blood decreases, suggesting that excessive pressure inhibits blood flow, which, in turn, results in a decrease in blood volume.

When the measurement tool is pressed strongly against the finger, pressure is applied to the capillaries and arterioles, and the blood vessels are compressed, impeding blood flow and increasing the photoplethysmogram’s DC component. Blood (whose flow is blocked) flows to the surrounding capillaries that are not pressed; consequently, the blood pressure increases in the compressed capillaries, and the blood flow is presumed to decrease. Therefore, it is thought that 1/FWHM_VPG_ (green light) behaves more like blood pressure than blood flow in capillaries.

Additionally, *a*/*S* (green light), (*a-b*)/(*a*-d) (green light), and 1/*ab* time (green light) exhibited the same tendency as 1/FWHM_VPG_ (green light).

The pulse transit time is the time difference between two distant points in a large artery. The Bramwell–Hill formula [18] represents PWV, which can be expressed as follows:(1)$$PWV^{2}=\left(\frac{∆P}{\rho }\right)\cdot \left(\frac{V}{∆V}\right).$$

Equation (1) shows that *PWV^2^* is proportional to the pressure change (pulse pressure, Δ*P*) and vascular volume (*V*) and inversely proportional to the blood density (*ρ*) and vascular volume change (Δ*V*). Equation (1) can be applied to large arteries. However, assuming that the time delay between the near-infrared light *a*-wave and the green light *a*-wave peaks can be regarded as the pulse wave transit time from the arteriole to the capillary, the above equation can be applied. The decrease in PWV (increase in pulse wave transit time) can be caused by a low pulse pressure in the path (arteriole to capillaries) or a soft blood vessel (large Δ*V/V*). It is unlikely that the blood vessels in patients with diabetes are soft; therefore, it can be assumed that the pulse pressure in the path (arteriole to capillary) is low. A low pulse pressure is equivalent to a low systolic blood pressure; consequently, it can be speculated that the low blood pressure in the arterioles and capillaries is the cause of the long delay from the near-infrared *a*-wave peak to the green *a*-wave peak.

However, peripheral hemodynamics have not yet been quantitatively evaluated. Because it can be assumed that peripheral hemodynamics are affected by the height of the heart, room temperature, noise, odor, fatigue, and sleep deprivation, in addition to medical history, sex, and age, the long-term measurement of fluctuations is considered more important than a single measurement.

## 5. Conclusions

In this study, we confirmed that there were clear differences between the group with diabetes and the control group regarding 1/FWHM_VPG_ (green light), which could be easily calculated using the volume photoplethysmography method and the time delay between the green and near-infrared light *a*-wave peaks.

We confirmed that the group with diabetes tended to have a small 1/FWHM_VPG_ (green light) and an extended time delay between the green and near-infrared light *a*-wave peaks. The speculated mechanisms underlying these results are as follows. Increased vascular resistance in capillaries and arterioles in patients with diabetes reduces the blood pressure in the capillaries and arterioles, which in turn lowers the 1/FWHM_VPG_ (green light). Reduced blood pressure in the capillaries and arterioles decreases PWV, which in turn increases the time delay between the green and near-infrared light *a*-wave peaks.

The photoplethysmography sensors used in this study can be integrated into low-cost, miniaturized, ring-type wearable devices, enabling continuous monitoring of the hemodynamics of arterioles and capillaries at home, which was previously challenging. Moreover, the proposed method could be useful in outpatient diagnosis, as well as in disease prevention. However, the extent to which poor long-term peripheral hemodynamics affect health remains unclear, and further research is required in this area.

## Figures and Tables

**Figure 1 sensors-24-07882-f001:**
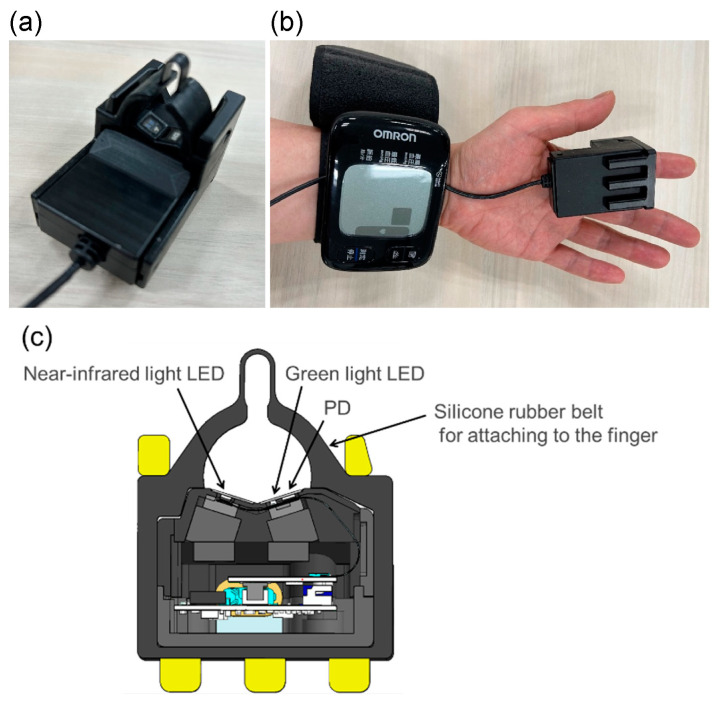
(**a**) Finger-mounted measurement tool; (**b**) mounted state; (**c**) cross-sectional view of the finger-mounted measurement tool.

**Figure 2 sensors-24-07882-f002:**
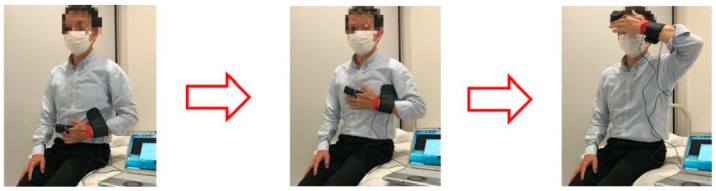
Measurement posture with the measurement device held at navel, chest, and forehead heights.

**Figure 3 sensors-24-07882-f003:**
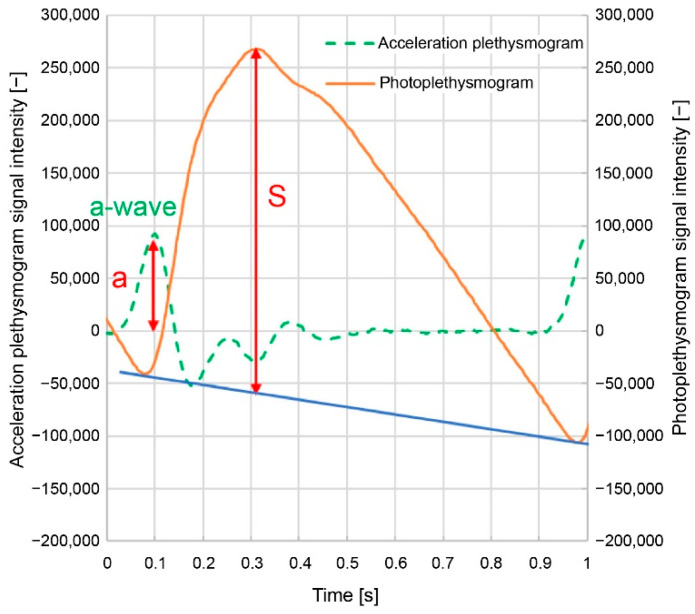
Photoplethysmogram height (*S*) and acceleration plethysmogram *a*-wave peak height (*a*).

**Figure 4 sensors-24-07882-f004:**
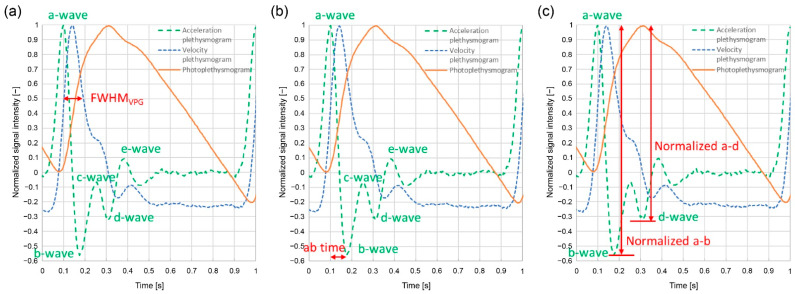
Velocity and acceleration plethysmogram feature values: (**a**) inverse full width at half-maximum velocity plethysmogram (FWHM_VPG_); (**b**) *ab* time; (**c**) normalized *a-b*, normalized *a-d*.

**Figure 5 sensors-24-07882-f005:**
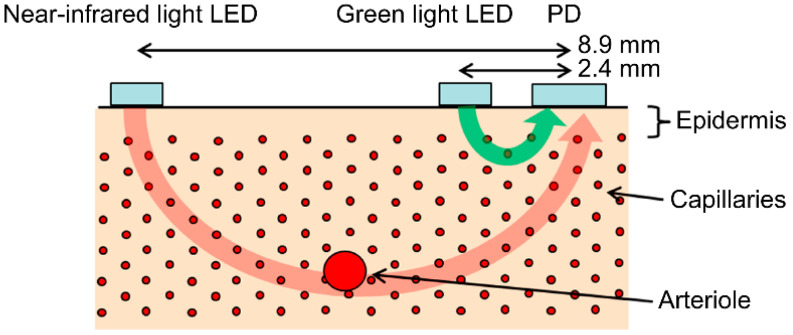
Penetration depth of green light (green arrow) and near-infrared light (red arrow) received by the photodiode (PD).

**Figure 6 sensors-24-07882-f006:**
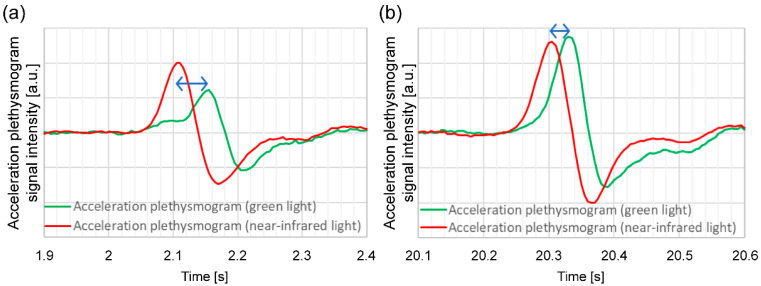
Example of changes in the *a*-wave peak time difference between green light and near-infrared acceleration plethysmograms (time delay between green light and near-infrared light *a*-wave peaks) for the same subject when the time delay (blue arrow) is (**a**) large and (**b**) small.

**Figure 7 sensors-24-07882-f007:**
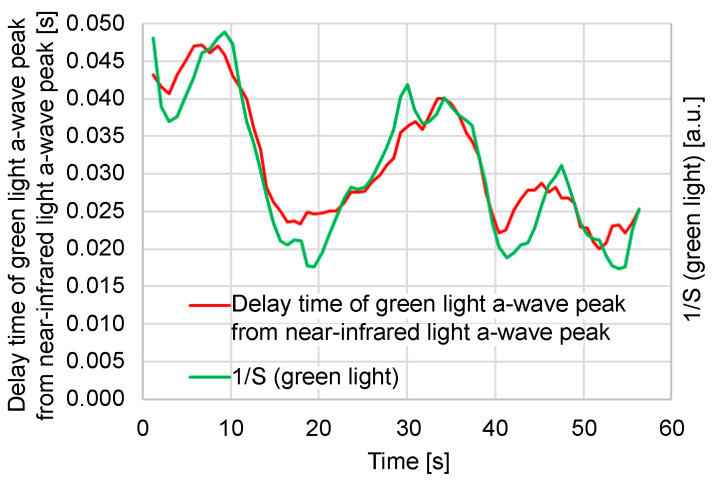
Relationship between time delay of green light *a*-wave peak from near-infrared light *a*-wave peak and reciprocal of green light photoplethysmogram amplitude (*S*).

**Figure 8 sensors-24-07882-f008:**
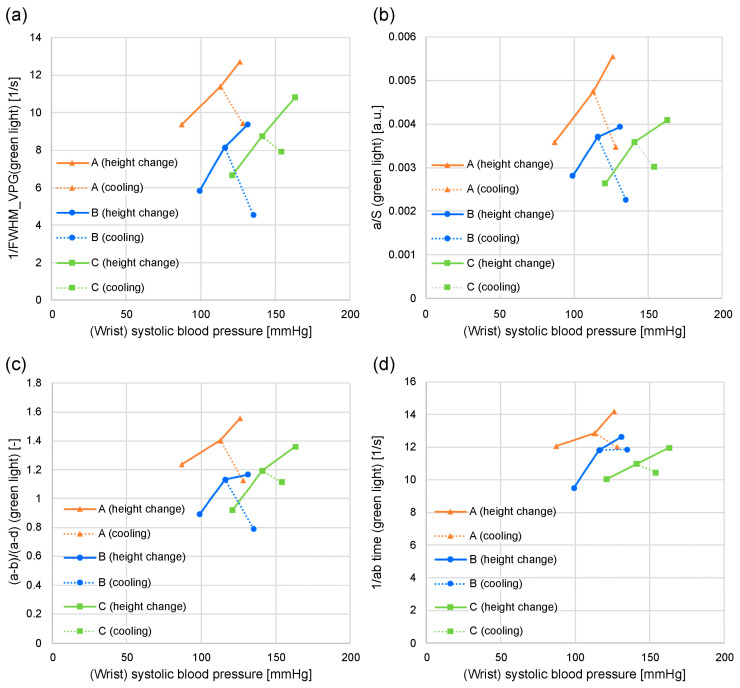
Relationship between systolic blood pressure and each feature value when the height of the measurement site (finger) from the heart changes and when the vicinity of the measurement site is cooled: (**a**) 1/FWHM_VPG_ (green light); (**b**) *a/S* (green light); (**c**) (*a-b*)/(*a-d*) (green light); (**d**) 1/*ab* time (green light).

**Figure 9 sensors-24-07882-f009:**
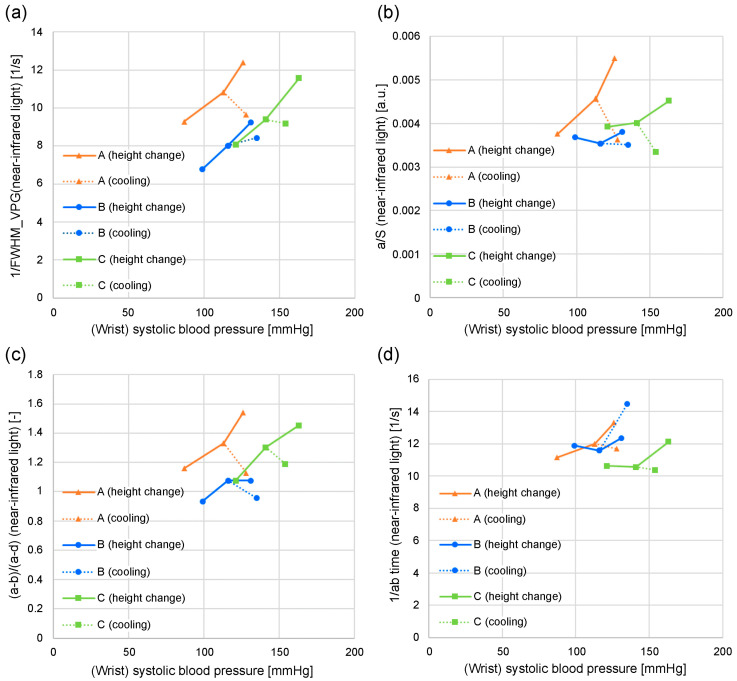
Relationship between systolic blood pressure and each feature value when the height of the measurement site (finger) from the heart changes and when the vicinity of the measurement site is cooled: (**a**) 1/FWHM_VPG_ (near-infrared light); (**b**) *a/S* (near-infrared light); (**c**) (*a-b*)/(*a-d*) (near-infrared light); (**d**) 1/*ab* time (near-infrared light).

**Figure 10 sensors-24-07882-f010:**
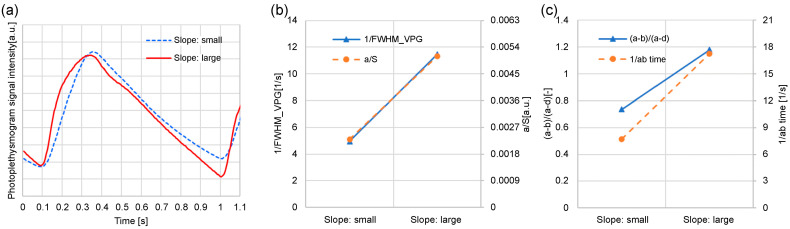
Examples of waveforms with different photoplethysmogram rising slopes: (**a**) photoplethysmogram waveform; (**b**) 1/FWHM_VPG_ and *a/S* change; (**c**) (*a-b*)/(*a-d*) and 1/*ab* time change.

**Figure 11 sensors-24-07882-f011:**
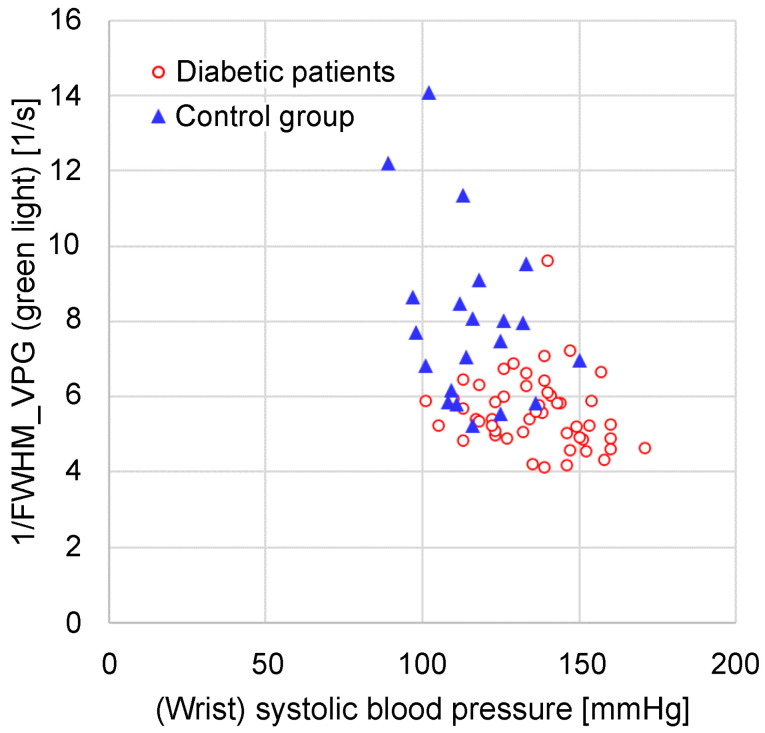
Relationship between wrist systolic blood pressure and 1/FWHM_VPG_ (green light).

**Figure 12 sensors-24-07882-f012:**
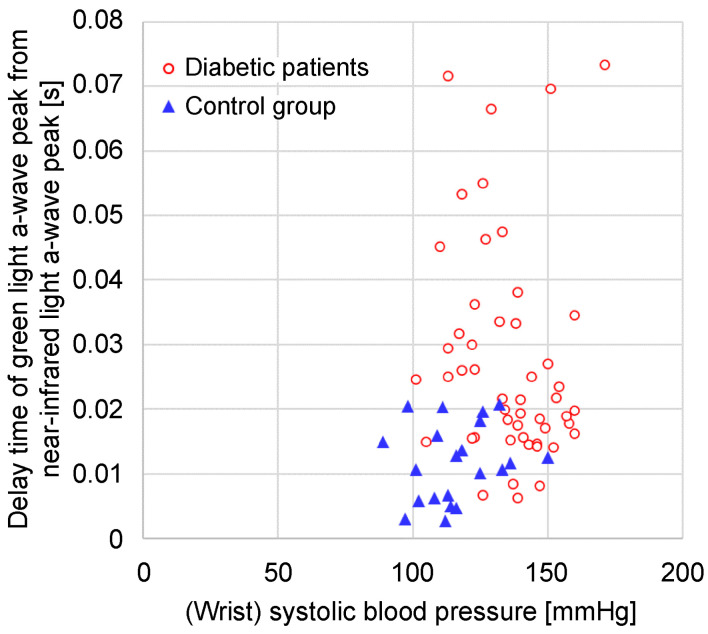
Relationship between the wrist systolic blood pressure and the time delay of the green light *a*-wave peak from the near-infrared light *a*-wave peak.

**Figure 13 sensors-24-07882-f013:**
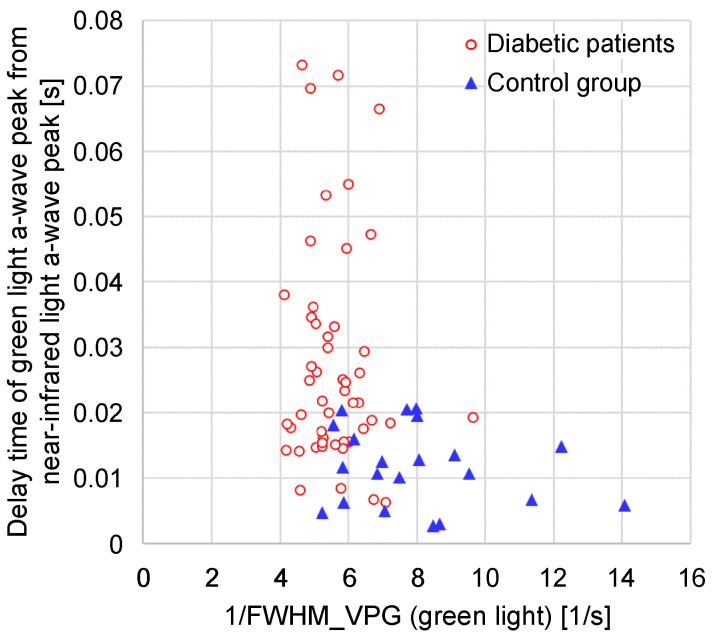
Relationship between 1/FWHM_VPG_ (green light) and time delay of green light *a*-wave peak from near-infrared light *a*-wave peak.

**Figure 14 sensors-24-07882-f014:**
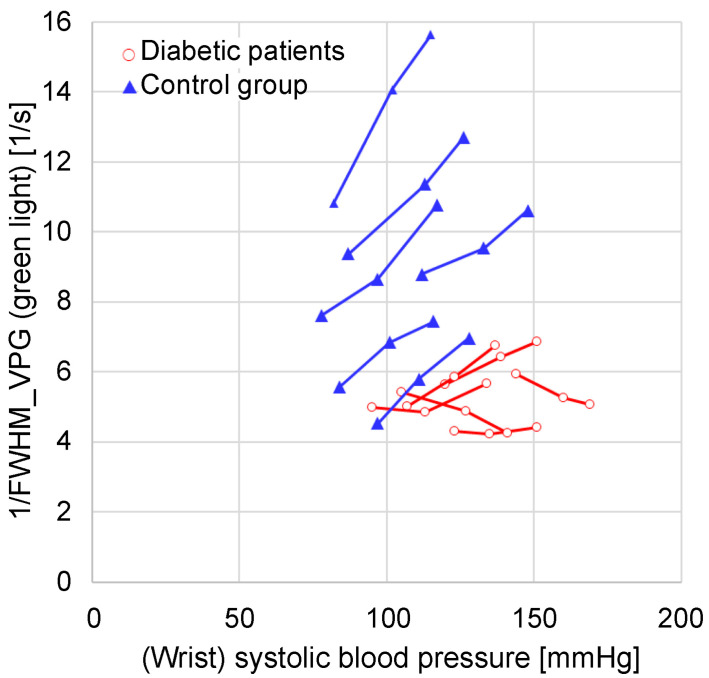
Relationship between wrist systolic blood pressure and 1/FWHM_VPG_ (green light) at various heights above the heart.

**Figure 15 sensors-24-07882-f015:**
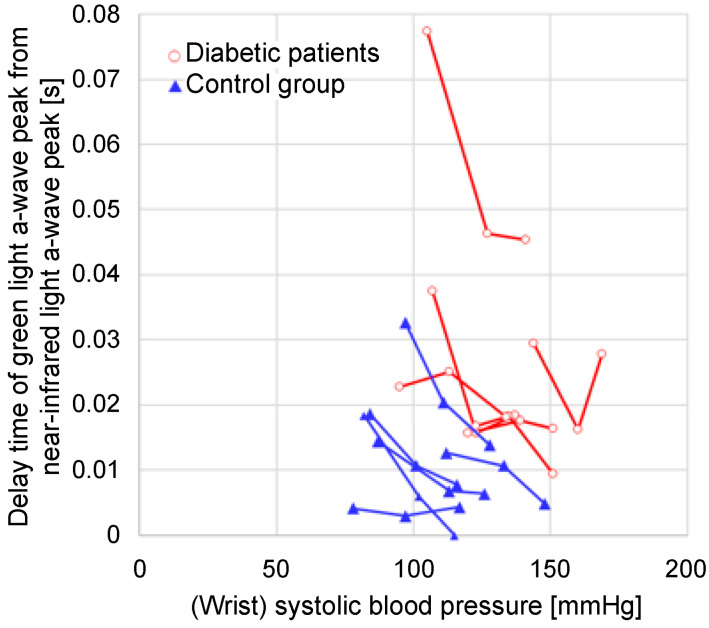
Relationship between wrist systolic blood pressure and time delay of green light *a*-wave peak from near-infrared light *a*-wave peak when the height above the heart changes.

**Figure 16 sensors-24-07882-f016:**
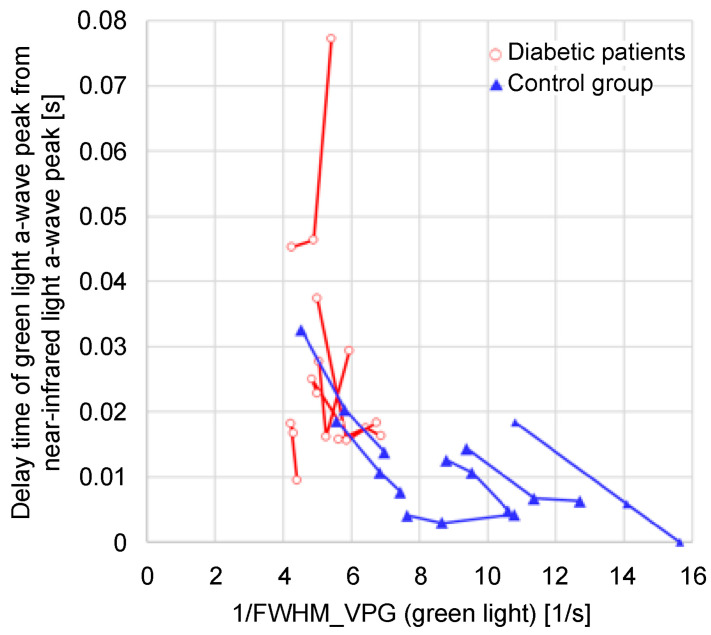
Relationship between 1/FWHM_VPG_ (green light) and time delay of green light *a*-wave peak from near-infrared light *a*-wave peak when the height from the heart changes.

**Figure 17 sensors-24-07882-f017:**
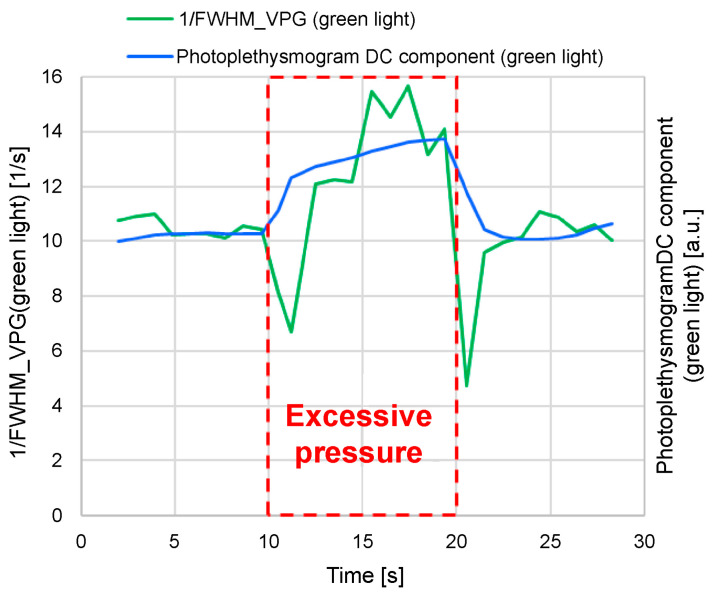
Photoplethysmogram direct-current component and 1/FWHM_VPG_ (green light) when excessive pressure is applied.

## Data Availability

Data are contained within the article.

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
