# Peer review of "Peripheral Hemodynamics Estimation Using the Photoplethysmography Method"

_sensors, 2024, doi:10.3390/s24247882_

Round 1

Reviewer 1 Report

Comments and Suggestions for Authors

I read this paper with high interest and authors provided a strategy using a skin sensors to detect peripheral hemodynamics. However, there exist scientific concerns in the LED laser source and detection depths. In addition, other suggestions have been attached as well.

1. the LED laser source and detection depths should be clearly explained as in Figure 5. For example, the rational depth of arteiole far away the skin top is between 0.3-1 mm. In this sense, the detection sensitivity of near infrared light LED is hard to reach some scales.

2. The schematic diagram of skin sensors in Figure 1 should be attached. For example, the locations of different sensors.

3. The programs applied in the signal process should be explained more.

Comments on the Quality of English Language

The quality of English language is okay for readers

Reviewer 2 Report

Comments and Suggestions for Authors

The manuscript used photoplethysmography method through green-light and near-infra-14 red LEDs to investigate peripheral hemodynamics on diabetic patients. The results showed that there were clear differences between the diabetic group and control group regarding related parameters in terms of peripheral hemodynamics. The study sounds interesting as highlighting the potential of a photoplethysmography device mounted on a ring-wearable device for diabetes monitoring. Overall, there are some major points be considered by the authors before it can be recommended for publication.

1. The section of materials and methods are confused with the part of results, i.e., some of the figures seems to be from the results rather than a description of methods. In short, the materials and methods should be presented concisely as for a better readership.

2. There are a lot of errors with grammar and punctuation throughout the manuscript. For example, in page 4 line 143, “The proposed design was such that…” “such” is redundant.; in page 15 line 413, “A It is evident that 1/FWHMVPG (green light) was small….”, “A” should be deleted. Thus, it is suggested to be reviewed by a native English speaker before it can be recommended for publication. To be honest, I had a bad time with reviewing the manuscript.

3. In figure 10a, what does it mean for slope small (dashed blue line)? should be clarified for a better understanding.

Comments on the Quality of English Language

It is suggested to be reviewed by a native English speaker before the manuscript can be recommended for publication. 

Round 2

Reviewer 1 Report

Comments and Suggestions for Authors

Agree to be published after minor correction, such as, to increase the pixel in Fig. 2

Reviewer 2 Report

Comments and Suggestions for Authors

The authors have considered and addressed all the comments raised in previous edition. I am glad to see that the revised manuscript is now suitable for publication in the present form in Sensors.